# Effect of Vacancy, As, and Sb Dopants on the Gold-Capturing Ability of Cu_2_S during Gold Collection in Matte Processes

**DOI:** 10.3390/molecules28217390

**Published:** 2023-11-02

**Authors:** Hui Huang, Huihui Xiong, Lei Gan

**Affiliations:** School of Metallurgy Engineering, Jiangxi University of Science and Technology, Ganzhou 341000, China; hh23n10y@hotmail.com

**Keywords:** Cu_2_S, As and Sb doping, gold collection in matte, adsorption mechanism

## Abstract

The technique of gold collection in matte can effectively improve the trapping efficiency of precious metals such as gold, silver, and platinum. However, the underlying mechanism of gold collection from high-temperature molten matte is complex and not well understood. In this work, the first-principle calculations were utilized to investigate the adsorption behavior of gold atoms on a Cu_2_S surface. The effects of vacancies and As and Sb doping on the gold-trapping ability of Cu_2_S were also explored, and the electronic properties of each adsorption system, including the charge density difference, density of states, and charge transfer, were systematically analyzed. The results show that the Cu-terminated Cu_2_S(111) surface has the lowest surface energy, and the Au atom is chemically adsorbed on the Cu_2_S(111) with an adsorption energy of −1.99 eV. The large adsorption strength is primarily ascribed to the strong hybridizations between Au-5d and Cu-3d orbitals. Additionally, the Cu vacancy can significantly weaken the adsorption strength of Cu_2_S(111) towards Au atoms, while the S vacancy can notably enhance it. Moreover, due to the formation of strong covalent As–Au/Sb–Au bonds, doping As and Sb into Cu_2_S(111) can enhance the gold-trapping capability of Cu_2_S, and the Sb doping exhibits superior effectiveness. Our studied results can provide theoretical guidance for improving the gold collection efficiency of Cu_2_S.

## 1. Introduction

The biological oxidation process [1,2] and fluidized roasting-cyanide leaching process are always employed to treat complex gold concentrates for the gold smelting industries [3,4]. However, these two technologies, although relatively mature, exhibit poor adaptability to raw materials and are ineffective in processing complex concentrates [5]. With the intensifying competition in the gold industry, there are more and more complex and difficult-to-process raw materials, such as As-containing, Sb-containing, or high-copper gold concentrates [6,7]. Although these concentrates can be pre-treated by both the biological oxidation process and roasting process, the ideal technical indexes cannot be achieved. To address this issue, the technique of gold collection in matte is proposed and applied in gold smelting industries due to its significant advantages, such as large processing scale, low cost, and strong raw-material adaptability. This technique not only enhances the recovery rate of valuable metals, including gold, silver, and copper but also facilitates the comprehensive recovery of rare and precious metals [8,9]. As is known, copper can serve as an effective collector of gold. By utilizing this property, the gold concentrates are effectively enriched from gangue and other base metals through the gold collection in the matte process. Subsequently, the individual metals can be separated to further improve the recovery rate of gold and other precious metals [10,11].

In recent years, extensive investigations have been conducted by scholars on the mechanism of gold collection from high-temperature molten matte [12,13,14,15,16,17]. For instance, AVARMA et al. [15,16] proposed that the distribution ratio of precious metals, such as Au and Ag, in the iron olivine slag and copper matte was determined by the properties of copper matte. During the smelting process, the precious metal elements are enriched in the copper matte by substituting the Cu or Fe of matte in the form of sulfides, and a high capturing rate is observed in high-grade copper matte. However, some scholars have claimed [18] that the underlying mechanism of gold collection in matte lies in the similar crystal structure and lattice parameters between gold and copper in the molten state, resulting in the formation of solid solutions or intermetallic compounds. Nevertheless, it should be noted that both atomic radii and lattice constants of Cu and Au are significantly different [19,20]. Moreover, the Cu_2_S in high-temperature molten matte exhibits a cubic structure, which is distinct from the face-centered cubic structure of gold, making it difficult for them to form solid solutions. Additionally, Guo et al. [11] argue that the multiphase distribution behavior of gold is determined by the smelting system and its thermodynamic properties, and the distribution of Au and Ag into molten matte leads to a reduction in the total Gibbs free energy of the system. In summary, the mechanism of Au captured in high-temperature smelting processes is complex, and there exists a lack of unanimity among scholars, highlighting the need for additional research in this area.

Previous experimental studies have revealed that high-temperature copper matte exhibits a good capturing ability towards gold, and the gold adsorption by copper matte should take place during the initial stage. Currently, the first-principle calculations based on the density functional theory (DFT) method can be employed to investigate the adsorption behaviors of metal atoms on solid surfaces at an atomic level [21,22,23,24]. Therefore, this work used the DFT computations to explore the adsorption characteristics of gold atoms on the Cu_2_S(111) surface and analyzed the microscopic interactions between the gold atoms and Cu_2_S(111) by calculating the electronic properties of each adsorption system, including the density of states, charge density, and charge density difference. Furthermore, the influence of vacancy defects on the gold adsorption of Cu_2_S(111) was also investigated. Lastly, as the gold-containing raw materials often contain antimony and arsenic, which tend to enter the copper matte, it is essential to investigate the adsorption of gold atoms on the Cu_2_S(111) surface doped with As and Sb. In conclusion, this paper comprehensively explored the influence of vacancies and As/Sb doping on the adsorption strength of Cu_2_S towards Au atoms, aiming to reveal the microscopic mechanism of gold collection in matte processes. Our studied results can provide guidance for practitioners to improve the process of the gold-trapping capability of Cu_2_S.

## 2. Results and Discussion

### 2.1. Adsorption of Au Atom on Pristine Cu_2_S(111)

The Cu_2_S in high-temperature molten matte is a cubic structure with a space group of FM-3M, as shown in Figure 1a. Each unit cell contains four S atoms and eight Cu atoms. The calculated lattice constant in this work is 5.616 Å, which agrees well with previous computational and experimental values [25]. Firstly, the most stable surface should be determined by comparing the surface energy (γ_s_) of different terminated surfaces; the γ_s_ can be calculated by [26,27]
(1)$$\gamma_{s}\approx [E_{\mathrm{slab}}\left(N\right)-NE_{\mathrm{bulk}}]/(2A)$$
where $E_{\mathrm{slab}}\left(N\right)$ is the total energy of the surface supercell, N is the number of atoms (or molecular formula) in the supercell, $E_{\mathrm{bulk}}$ represents the energy of each atom (or molecular formula) in the bulk material, and A is the corresponding surface area. There are three high-symmetry low-index surfaces for the Cu_2_S, namely (100), (110) and (111). Among them, both Cu_2_S(100) and Cu_2_S(111) have two distinct terminations, namely Cu-termination and S-termination, as illustrated in Figure 1. While the Cu_2_S(110) only has one kind of surface, namely the Cu/S-end surface. The calculated surface energies of Cu-and S-terminated Cu_2_S(100) are 0.99 J/m^2^ and 0.81 J/m^2^, respectively. And the surface energy of Cu_2_S(110) is 0.51 J/m^2^, while the surface energies of Cu-and S-terminated Cu_2_S(111) are 0.35 J/m^2^ and 0.91 J/m^2^, respectively. Generally, the lower surface energy for a surface means its higher stability. Notably, the Cu-terminated Cu_2_S(111) exhibits the lowest surface energy, suggesting that it has the highest exposure probability during the high-temperature smelting process. Consequently, the subsequent study only focuses on the adsorption behavior of Au atoms on the Cu-terminated Cu_2_S(111) surface.

Next, the 2 × 2 × 1 supercell of Cu_2_S(111) with 15 atomic layers is constructed to explore the adsorption behavior of the Au atom on the surface, as shown in Figure 2a. During the simulation process, the six-layer atoms at the bottom of the slab are fixed to mimic the bulk phase, while the remaining atoms and gold atoms are allowed to be relaxed. After full relaxation, the topmost Cu atoms and the second-layer S atoms are found to be located in the same plane (Figure 2b). It can be observed that there exist six possible adsorption sites for the Cu_2_S(111), namely the Cutop site, Stop site, B_Cu-S_ site, B_S-S_ site, B_Cu-Cu_ site, and hollow site. For the adsorption process, the gold atoms are placed above the adsorption sites with a height of 2.5 Å, and then the initial structures are subjected to fully relaxed. The calculated adsorption energies of different systems are listed in Table 1. One can see that the adsorption energies of Au atoms on the Cu_2_S(111) range from −1.54 eV to −1.99 eV. The Au atom on the hollow site is moved on the top of B_Cu-Cu_ after full relaxation, and the adsorption energy (−1.99 eV) at the B_Cu-Cu_ site is higher than those of other adsorption sites. Therefore, the Au atom is preferentially adsorbed onto the B_Cu-Cu_ site. As shown in Figure 3, the distances between the Au atom and its two nearest copper atoms are 2.594 Å and 2.591 Å, respectively, both of which are smaller than the radii sum of Au and Cu atoms (r_Au_ + r_Cu_ = 3.190 Å). Therefore, it can be inferred that the Au atom is chemically adsorbed onto the Cu_2_S(111) surface due to the large adsorption strength and small bond length.

In order to gain a deeper understanding of the interactions between the Au atom and Cu_2_S(111) surface, the charge density difference (CDD) and electron density distribution (EDD) of the most stable system are calculated, and the results are displayed in Figure 4. From Figure 4a,b, there are significant electron accumulation regions around the gold atom, while substantial electron depletion regions surround the two neighboring Cu atoms. This suggests that the partial electrons of Cu atoms are transferred to the Au atom, resulting in the formation of a strong Au–Cu metallic bond. Additionally, as depicted in Figure 4c, an abundance of electrons can be observed between the Au atom and its two neighboring Cu atoms, which indicates that these electrons are shared by them. This implies the Au–Cu metallic bond exhibits covalent characteristics, demonstrating that the strong interactions of Au and Cu atoms take place in the adsorption system.

Figure 5 gives the density of states of the most stable adsorbed Cu_2_S(111) with an Au atom, where Cu1 and Cu2 represent the two nearest-neighbor Cu atoms of the Au atom, as shown in Figure 4a. One can see that there are strong interactions between the Au atom and Cu1 and Cu2 atoms, which is primarily attributed to the hybridizations between the Au-5d and Cu-3d orbitals. Specifically, the Au-5d orbital obviously overlaps with the Cu1-3d orbitals in the energy range of −3.94 eV to −0.38 eV, and two significant resonance peaks can be found at about −2.18 eV and −1.63 eV, resulting in the formation of strong Au–Cu metallic bonding. Furthermore, there is a relatively weak interaction between the Au-6s and Cu1-3d orbitals in the energy range of −1.90 eV to 0.40 eV, and a resonant peak appears at the Fermi level. In contrast, the orbital hybridization between the Au-5p and Au-sp remains minimal across the entire energy level, leading to their negligible interaction.

### 2.2. Effect of the Vacancy on the Au Adsorption of Cu_2_S(111)

During the high-temperature melting process, some atoms can acquire enough energy to break free from the surrounding atoms and create vacancies on the Cu_2_S(111) surface. These vacancies have a significant impact on the redistribution of charge density and subsequently affect the adsorption of gold atoms. Therefore, it is necessary to investigate the effect of vacancies on the adsorption of Au atoms on the Cu_2_S(111). As displayed in Figure 6a–c, there are two types of vacancy defects on the Cu_2_S(111) surface, namely Cu vacancy and S vacancy. Based on this, the adsorption energies of Au atoms on the Cu_2_S(111) surface with Cu vacancy (Cu_2_S(111)-Cuvac) and S vacancy (Cu_2_S(111)-Svac) are calculated, and the most stable adsorption structures are depicted in Figure 6d–g. The adsorption energies of Au atoms on Cu_2_S(111)-Cuvac and Cu_2_S(111)-Svac are −0.91 eV and −3.04 eV, respectively. Compared with the Au@Cu_2_S(111) system (*E*_ads_ = −1.99 eV), the presence of Cu vacancy can significantly weaken the adsorption of Au atom on Cu_2_S(111). Conversely, the S vacancy can dramatically enhance the adsorption strength. Therefore, it can be inferred that the S vacancies on the Cu_2_S(111) surface can enhance the gold-capturing efficiency of Cu_2_S.

Figure 7 shows the CDD plots of Au@Cu_2_S(111)-Cuvac and Au@Cu_2_S(111)-Svac systems. In the Cu vacancy system, the Au atom gains about 0.013 e, as proved by the red region around it. In contrast, the three neighboring S atoms all experience a charge loss of 0.069 e. This indicates the partial electrons of S atoms are transferred to the Au atom, leading to the formation of a weak Au–S covalent bond. In the S vacancy system (Figure 7b), the abundant electron accumulation regions can be observed around the Au and Cu atoms, while the electron depletion regions are accumulated around the S atoms. This suggests that some electrons of S atoms are also transferred to the Au and Cu atoms, with the Au atom gaining about 0.131 e. Thus, the charge transfer number of Au in the Au@Cu_2_S(111)-Svac system is significantly greater than that in the Au@Cu_2_S(111)-Cuvac system, which leads to the stronger interaction between Au atom and Cu_2_S(111)-Svac.

The charge density distributions of the two adsorption systems are displayed in Figure 8. In the Au@Cu_2_S(111)-Cuvac system (Figure 8a), the Au atom shares some charges with the first-layer S atoms, resulting in the formation of a strong Au–S covalent bond. However, the Au atom only shares a small number of electrons with the Cu atoms in the subsurface, leading to the formation of weak Au–Cu metallic bonds. In contrast, in the Au@Cu_2_S(111)-Svac system (Figure 8b), the adsorbed Au atom shares a large amount of electrons with the first-layer Cu atoms and shares massive electrons with the second-layer Cu atoms, which results in the formation of two strong Au–Cu metallic bonds. This explains why the Cu_2_S(111)-Svac exhibits the stronger adsorption strength towards the Au atom.

The density of states (DOSs) of Au@Cu_2_S(111)-Cuvac and Au@Cu_2_S(111)-Svac systems were also calculated and shown in Figure 9. From Figure 9a, the obvious orbital hybridizations between Au-d and S-p can be found in the energy range of −7.50 to 0.00 eV, and there is a slight orbital overlap between Au-s and S-p within the range of 0.00 to 1.35 eV. In addition, the weak orbital hybridizations between Au-d and Cu-d can be found in the energy ranging from −7.50 to 0.00 eV. However, none of these hybridizations results in prominent resonance peaks. Thus, the Cu_2_S(111)-Cuvac exhibits the weak adsorption strength of the Au atom. For the Au@Cu_2_S(111)-Svac system, as shown in Figure 9b, there is an obvious overlap between the Au-d orbital and d orbital of the first layer Cu atom in the energy range of −7.50 eV to −0.50 eV, and two distinct resonance peaks can be observed at approximately −4.80 eV and −2.70 eV, which results in the formation of strong Au–Cu metallic bonds. Furthermore, the significant hybridizations between the Au-d orbital and d orbital of the second layer Cu atom can also be found within the energy range of −7.50 eV to 0.00 eV, leading to the formation of a strong Au–Cu metallic bond. Therefore, the Cu_2_S(111)-Svac exhibits strong adsorption capacity for Au atom.

### 2.3. Effect of As and Sb Dopant on the Au Adsorption of Cu_2_S

Currently, the presence of complex and challenging gold-containing raw materials has become increasingly prevalent, resulting in the inevitable ingress of As and Sb into copper matte. Therefore, it is necessary to study the adsorption of Au atom on the Cu_2_S(111) with As and Sb doping. As illustrated in Figure 10, the As and Sb atoms have the potential to be incorporated into Cu-site or S-site of Cu_2_S(111). To determine their preferred substitution sites, the binding energy (*E*_bin_) of different doping systems is determined by [28,29]
(2)$$E_{\mathrm{bin}}=E_{\mathrm{As}/\mathrm{Sb}-\mathrm{Cu}2S111}-E_{\mathrm{Cu}2S111-\mathrm{vac}}-E_{\mathrm{As}/\mathrm{Sb}}$$

In this equation, $E_{\mathrm{As}/\mathrm{Sb}-\mathrm{Cu}2S111}$ is the total energy of As/Sb-doped Cu_2_S(111), $E_{\mathrm{Cu}2S111-\mathrm{vac}}$ is the total energy of Cu_2_S(111) with a Cu or S vacancy, and the $E_{\mathrm{As}/\mathrm{Sb}}$ is the energy of a single As/Sb atom in its bulk phase. It should be noted that a system with a lower binding energy tends to be more stable. Herein, the binding energies of various doped systems were calculated and displayed in Figure 10c. One can see that the binding energies at the S-site are significantly lower than those at the Cu-site for both As- and Sb-doped systems. Therefore, the doping of As/Sb at the S-site exhibits superior stability, and its stability is larger than that of clean Cu_2_S(111) due to the smaller binding energy. Consequently, the subsequent investigations only focus on the Cu_2_S(111) with As/Sb doped at the S-site.

For the As/Sb-doped Cu_2_S(111), there exist five possible adsorption sites, including the Astop site, Cutop site, B_Cu-As_ site, B_As-S_ site, and hollow site, as shown in Figure 10b. By performing geometric optimization and comparing the *E*_ads_ of different adsorption systems, the two most stable adsorption structures are obtained (Figure 11). The atomic structures of As- and Sb-doped Cu_2_S(111) undergo significant deformation after Au adsorption, and the Au atom is preferentially adsorbed on the B_Cu-As_ or B_Cu-Sb_ site. Furthermore, the adsorption energies of Au atoms on As- and Sb-doped Cu_2_S(111) are −2.11 eV and −2.28 eV, respectively, both of which are higher than the that of Au atoms on pure Cu_2_S(111). Therefore, we can infer that both As and Sb doping can enhance the gold-capturing ability of Cu_2_S, with Sb doping having a more pronounced effect. Additionally, there are abundant yellow regions between As/Sb and Au atoms, as plotted in Figure 11b,d, illustrating that the substantial electrons are shared by them, and the As–Cu or Sb–Au covalent bond is formed. Moreover, the adsorbed Au atoms in the two systems also share some electrons with the two Cu atoms in the second layer, implying that the Au–Cu metallic bonds are formed. Therefore, the As/Sb-doped Cu_2_S(111) exhibits strong adsorption capabilities for gold atoms.

Figure 12 displays the DOSs and CDD of the most stable Au@As/Sb-doped Cu_2_S(111). From Figure 12a, the Au-d orbital undergoes significant hybridization with As-p orbital in the energy range of −7.50 eV to 2.50 eV and exhibits a weak interaction with s, d orbitals of As atom. Moreover, a prominent resonance peak is observed at about −3.60 eV. And there is a strong orbital overlap between Au-d and Cu-d in the energy range of −7.50 eV to 0.00 eV. Further, the abundant charges are accumulated between the Au and As atoms, as depicted by the CDD of Figure 12c. Specifically, the Au atom gains approximately 0.098 e, while the As atom loses about 0.109 e, resulting in the formation of a strong As–Au covalent bond. Meanwhile, an amount of charge is present between the Au and two Cu atoms in the second layer, demonstrating that the Au–Cu metallic bond is formed.

When the Au atom is adsorbed on the Sb-doped Cu_2_S(111) (Figure 12b), the Au-d orbital strongly interacts with the Sb-p orbital in the energy range of −7.50 eV to 2.50 eV, and a small resonant peak can be observed at around −9.30 eV for the Au-s and Sb-s orbitals, resulting in the strong interactions between Au and Sb atoms. Moreover, a significant orbital overlap exists between the Au-d and Cu-d orbitals, ranging from −7.50 eV to 0.00 eV. As shown in Figure 12d, the Au atom gains approximately 0.147 e, while the Sb atom loses about 0.121 e. The larger charge transfer number between Au and Sb atoms indicates the Au–Sb covalent bond has a larger strength in comparison with the Au–As covalent bond. Similarly, the Au–Cu metallic bond is also formed between the Au and two Cu atoms in the second layer. Thus, the adsorption strength of the Au atom on Sb-doped Cu_2_S(111) is greater than that on As-doped Cu_2_S(111).

In summary, the initial stage of gold-capturing by copper matte is an adsorption process [11], and a higher adsorption energy of gold on the Cu_2_S surface means a higher gold-capturing efficiency. Based on the above discussion, the clean Cu_2_S(111) possesses a superior ability to capture gold due to its large adsorption energy. And the S vacancy on the Cu_2_S(111) enhances the gold-trapping capability, while the Cu vacancy reduces the gold-capturing efficacy. Therefore, the practitioners should focus on controlling the actual operation technique to increase the number of S vacancies. Additionally, doping As/Sb into the S site of Cu_2_S(111) significantly improves the gold-trapping capability of Cu_2_S, which implies that the presence of arsenic and antimony compounds in the raw materials is beneficial to the gold collection in the matte process. These results can provide practical guidance for practitioners to improve the process of the gold-trapping efficacy of Cu_2_S and also pave the way for the comprehensive utilization of arsenic and antimony-containing gold concentrates.

## 3. Calculation Method and Details

All calculations in this work were performed using the DMol^3^ package [30] within the Materials Studio 2019 software. The Perdew–Burke–Ernzerh (PBE) functional of generalized gradient approximation (GGA) was chosen as the exchange-correlation functional [31]. The DFT semicore pseudopotential (DSSP) method [32] was used to treat the interactions between core electrons and nucleus. The double numerical polarization (DNP) basis set [32] and a cutoff radius of 5.2 Å were adopted during the simulation process. For geometry optimization, the energy convergence tolerance, the maximum force, and the maximum displacement were set to 1.0 × 10^−5^ Ha, 2.0 × 10^−3^ Ha/Å and 5 × 10^−3^ Å, respectively. The self-consistent field (SCF) tolerance for convergence was set to 1.0 × 10^−5^ Ha. For the bulk Cu_2_S, the k-point meshes of 10 × 10 × 10 sampled in the Brillouin zone [33] were used. For the slab models, the k-points meshes of 5 × 5 × 1 and 10 × 10 × 1 were used for the geometry optimization and electronic property calculations, respectively. To account for the periodicity of the adsorption system, a vacuum layer of 15 Å was applied in the z-direction to eliminate the interactions between periodic structures.

The adsorption energy (*E*_ads_) can be used to evaluate the interaction strength between Au atom and Cu_2_S surface, which can be obtained by the following equation [24,34]:(3)$$E_{\mathrm{ads}}=E_{{\mathrm{Cu}}_{2}S/\mathrm{Au}}-E_{{\mathrm{Cu}}_{2}S}-E_{\mathrm{Au}}$$
where $E_{{\mathrm{Cu}}_{2}S/\mathrm{Au}}$ is the total energy of Au atom adsorbed on Cu_2_S surface, $E_{{\mathrm{Cu}}_{2}S}$ and $E_{\mathrm{Au}}$ are the energies of clean Cu_2_S surface and a single Au atom, respectively.

## 4. Conclusions

In this work, the DFT method was employed to investigate the adsorption behavior of the Au atom on the Cu_2_S surface, and the effects of Cu/S vacancy and As/Sb doping on the Au adsorption of Cu_2_S were also explored, which aims to reveal the micro-mechanism of gold collection in matte. The main conclusions are as follows:
(1)The Cu-terminated Cu_2_S(111) exhibits the lowest surface energy, resulting in its preferential exposure during the high-temperature smelting process.(2)Gold atom is preferentially adsorbed on the B_Cu-Cu_ site, with an adsorption energy of −1.99 eV. The strong chemisorption is mainly attributed to the hybridizations between Au-5d and Cu-3d orbitals.(3)The presence of Cu vacancy weakens the adsorption strength of the Au atom on the Cu_2_S(111), whereas the S vacancy notably enhances the adsorption strength. Thus, the S vacancy of Cu_2_S(111) can effectively improve the Au-capturing efficiency of Cu_2_S.(4)As/Sb atom preferentially substitutes for the S atom in the topmost layer of Cu_2_S(111). Doping As and Sb into the Cu_2_S(111) can enhance the Au adsorption and Au-capturing capabilities, and the Sb doping exhibits superior effectiveness.

## Figures and Tables

**Figure 1 molecules-28-07390-f001:**
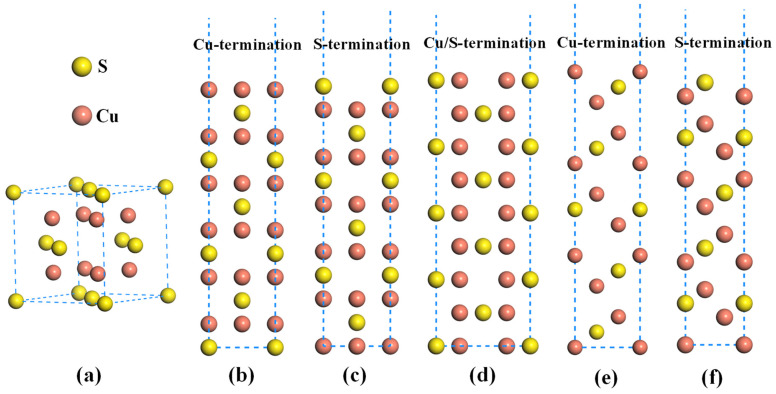
Atomic configurations of different structures: (**a**) bulk Cu_2_S, (**b**,**c**) Cu_2_S(100), (**d**) Cu_2_S(110) and (**e**,**f**) Cu_2_S(111).

**Figure 2 molecules-28-07390-f002:**
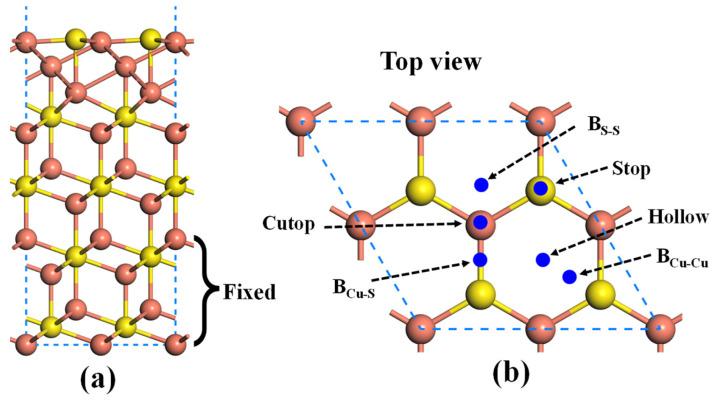
(**a**) Atomic structure of Cu-terminated Cu_2_S(111) and (**b**) its six possible adsorption sites. The orange and yellow balls represent the Cu and S atoms, respectively.

**Figure 3 molecules-28-07390-f003:**
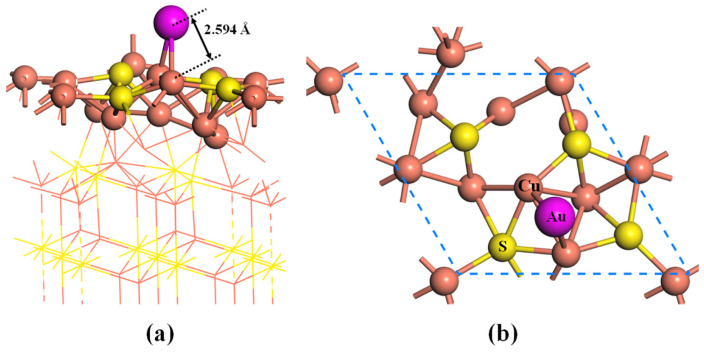
Side view (**a**) and top view (**b**) of the most stable structure of Au atom adsorbed on the Cu-terminated Cu_2_S(111). The orange, yellow and pink balls represent the Cu, S and Au atoms, respectively.

**Figure 4 molecules-28-07390-f004:**
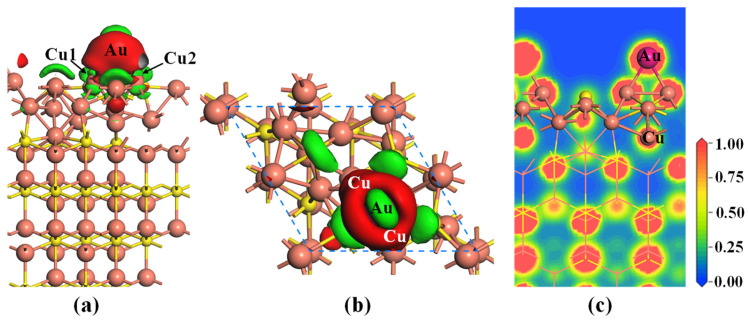
(**a**) Side view and (**b**) top view of charge density difference (CDD), and (**c**) electron density distribution (EDD, e/Å^3^) for the most adsorbed Cu_2_S(111) with an Au atom. The red and green regions in (**a**,**b**) represent the electron gains and losses, and the isovalue of CDD is ±0.005 e/Å^3^. The orange and yellow balls represent the Cu and S atoms, respectively.

**Figure 5 molecules-28-07390-f005:**
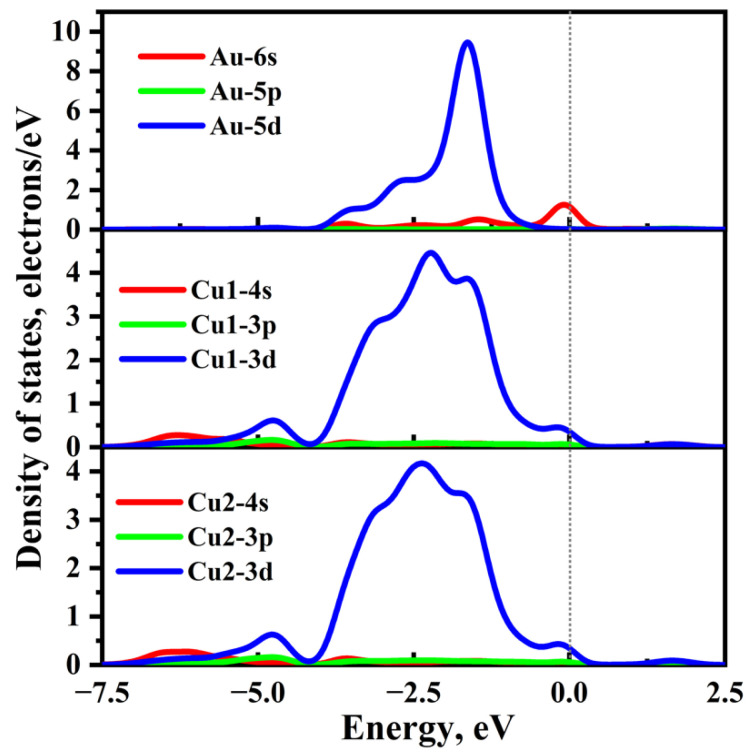
Density of states (DOSs) of the most stable adsorption system for the Au atom on Cu-terminated Cu_2_S(111), the Fermi level is set to zero.

**Figure 6 molecules-28-07390-f006:**
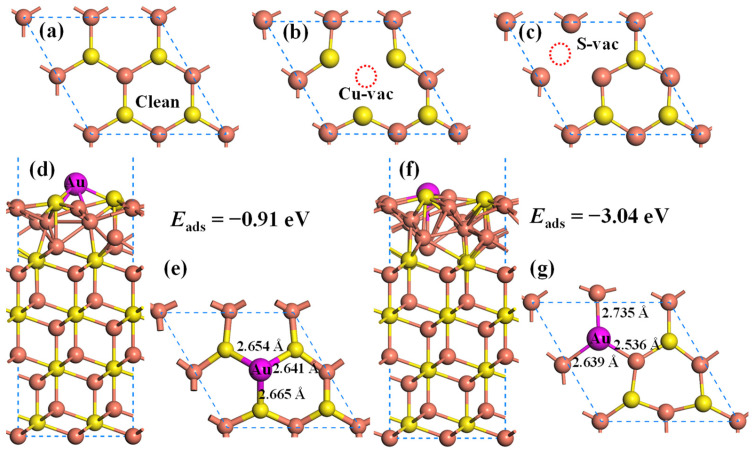
Atomic configurations of different structures. (**a**) clean Cu_2_S(111), (**b**) Cu_2_S(111) with Cu vacancy, (**c**) Cu_2_S(111) with S vacancy, (**d**,**e**) Au@ Cu_2_S(111)-Cuvac and (**f**,**g**) Au@ Cu_2_S(111)-Svac. The orange, yellow and pink balls represent the Cu, S and Au atoms, respectively.

**Figure 7 molecules-28-07390-f007:**
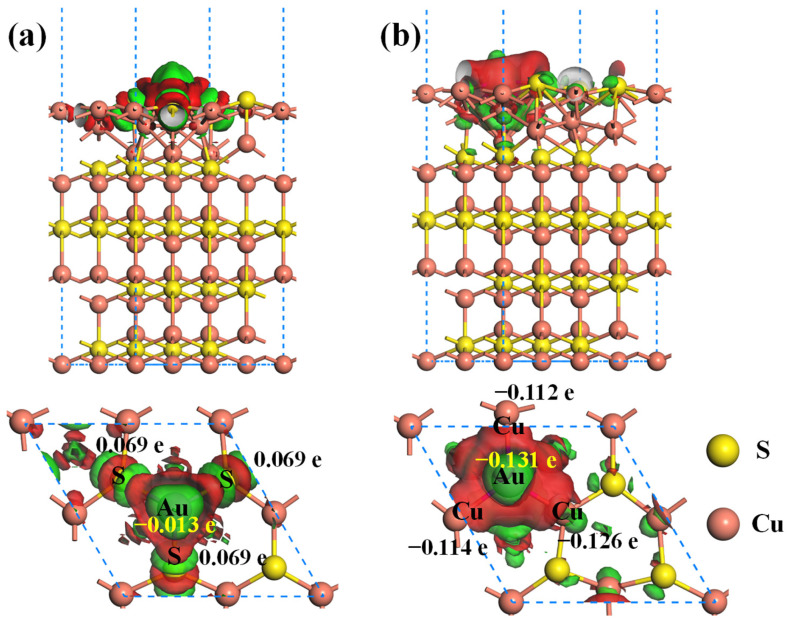
The CDD plots of (**a**) Au@Cu_2_S(111)-Cuvac and (**b**) Au@Cu_2_S(111)-Svac systems. The red and green regions represent the electron gains and losses, and the isovalue of CDD is ±0.005 e/Å^3^.

**Figure 8 molecules-28-07390-f008:**
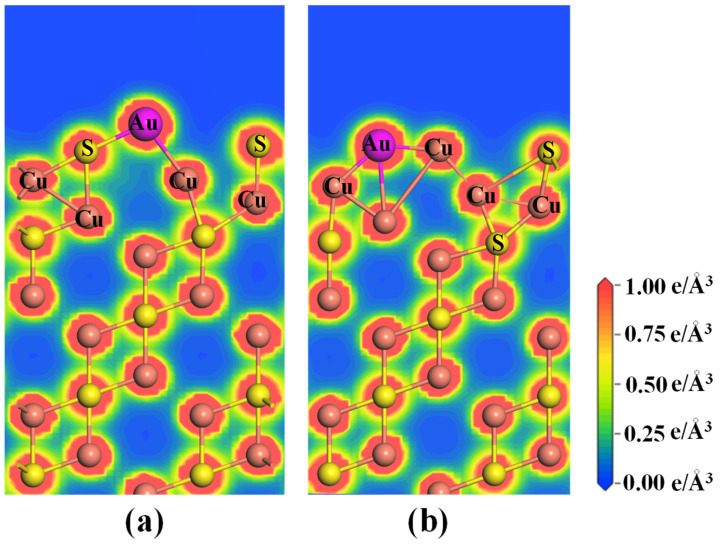
The CDD plots of (**a**) Au@ Cu_2_S(111)-Cuvac and (**b**) Au@ Cu_2_S(111)-Svac systems.

**Figure 9 molecules-28-07390-f009:**
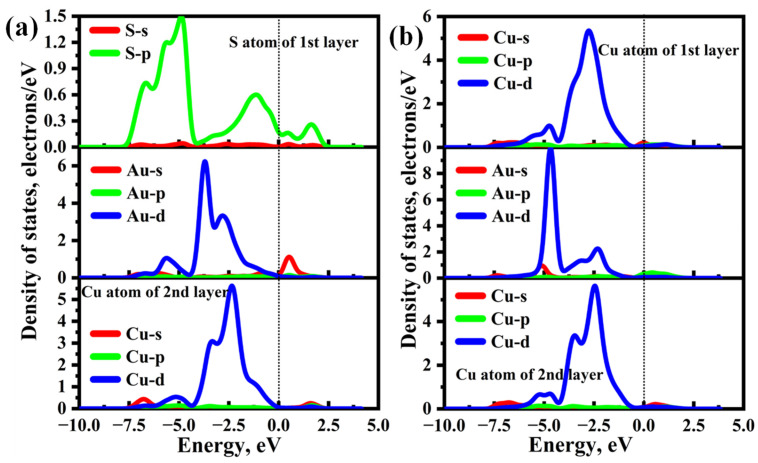
The density of states (DOSs) of (**a**) Au@ Cu_2_S(111)-Cuvac and (**b**) Au@ Cu_2_S(111)-Svac, and the dashed lines refer to the Fermi level.

**Figure 10 molecules-28-07390-f010:**
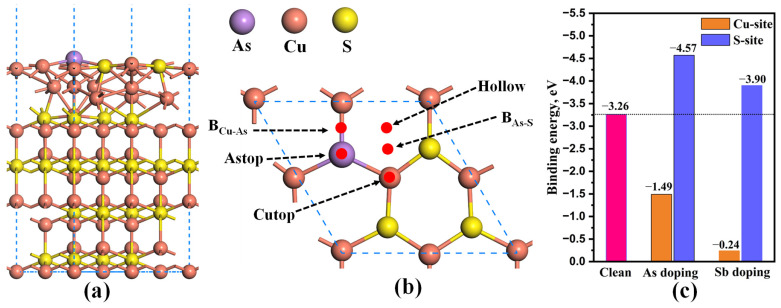
(**a**) Side view and (**b**) top view of As/Sb-doped Cu_2_S(111), and (**c**) the binding energies of clean Cu_2_S(111) surface (pink histogram) and As/Sb-doped systems.

**Figure 11 molecules-28-07390-f011:**
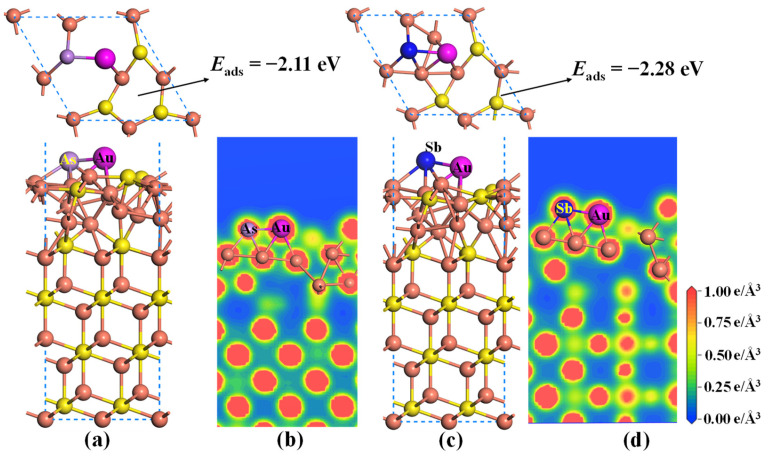
The lowest-energy structures of Au adsorbed on As/Sb-doped Cu_2_S(111) (**a**,**c**) and their corresponding charge density distributions (**b**,**d**). The orange and yellow balls represent the Cu and S atoms, respectively.

**Figure 12 molecules-28-07390-f012:**
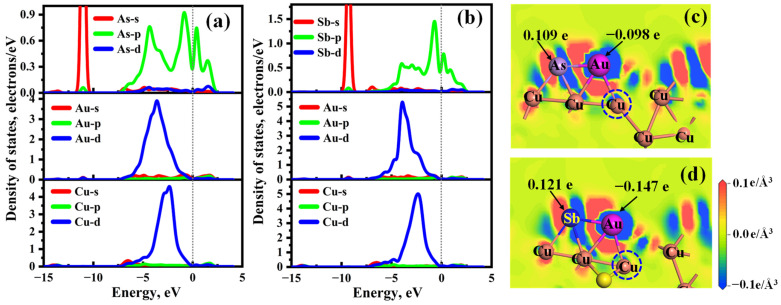
Density of states (**a**,**b**) and charge density difference (**c**,**d**) of Au adsorbed on As/Sb-doped Cu_2_S(111), the Cu atoms in (**a**,**b**) are labeled by the dashed circles in (**c**,**d**).

**Table 1 molecules-28-07390-t001:** Adsorption energies of Au atom on the Cu_2_S(111) with different adsorption sites.

Adsorption Sites	Stop	B_Cu-Cu_	B_Cu-S_	B_Cu-S_	Cutop	Hollow
*E* _ads_	−1.54	−1.99	−1.84	−1.81	−1.67	−1.99

## Data Availability

Not applicable.

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
