# Peer review of "Effect of Vacancy, As, and Sb Dopants on the Gold-Capturing Ability of Cu2S during Gold Collection in Matte Processes"

_molecules, 2023, doi:10.3390/molecules28217390_

Round 1
Reviewer 1 Report
Comments and Suggestions for Authors
The authors present a comprehensive theoretical examination of potential interaction sites between Au atoms and a Cu2S surface, both with and without the incorporation of As and Sb atoms into the surface. This study is well-structured and written, ultimately shedding light on the most relevant interaction sites. I only have one observation, on page four, line 154. In this instance, the authors assert that electrons are shared between the Au atom and the Cu atoms. Yet, in the subsequent sentence, they indicate that the Au-Cu bond exhibits ionic attributes. These two statements appear contradictory, as shared electrons typically imply covalent interactions, not ionic ones. To reconcile this disparity, it is recommended that the authors consider calculating the Laplacian of the electron density, a valuable index for distinguishing between closed-shell and shared interactions.
Reviewer 2 Report
Comments and Suggestions for Authors
The manuscript entitled “Effect of vacancy, As and Sb dopant on the gold capture ability of Cu2S during gold collection in matte process” reported a comprehensive and detailed simulation study to investigate the adsorption behavior of gold atom on the Cu2S surface, and the effects of vacancy, As and Sb doping. The authors have tested surface energies of 5 different surface terminations and found that Cu-terminated Cu2S(111) exhibits the lowest surface energy, which was then selected as the substrate for the subsequent study. By performing DFT simulations, electron density analysis and charge density analysis, it as concluded that Gold atom is preferentially adsorbed on the BCu-Cu site, S vacancy of Cu2S(111) can effectively improve the Au capture rate of Cu2S, and As and Sb dopants can enchance the Au adsorption and Au-capturing capabilities. I recommend this work should be reconsidered for publication after some minor modifications, and the following issues should be stressed:
1. The introduction part has discussed a lot about high temperature and smelting process, but the subsequent DFT studies did not contain simulations related to temperature factors.
2. In Figure 10(c), it is suggested that binding energies for original Cu2S surface without doping also added for better comparison.
3. The charge density plots in Figure 11(b, d) and Figure12(c, d), the unit of the scale bars should be provided.
Comments on the Quality of English LanguageMinor editing of English language required
Reviewer 3 Report
Comments and Suggestions for Authors
The authors, by employing theoretical (DFT) evaluations, have studied the adsorption behavior of gold atom on the Cu2S surface. The effects of vacancies, As and Sb doping on the gold capturing ability of Cu2S have been also investigated and an analysis of the charge density difference, density of states and charge transfer has been provided. This is a good piece of work. The paper is timely and will be of interest to researchers and practitioners working in the field of precious metal extraction and metallurgy. The calculations have been executed with great care and deep understanding of the features of both the theoretical approach employed and the system under study. Results obtained are convincing. Paper is well structured and written. I recommend publication. One thing: The theoretical conclusions indeed may serve as guidelines for practitioners in improving the process of gold trapping efficacy of Cu2S. However, this is not very clearly outlined in the manuscript and I suggest elaborating on this matter and emphasizing on the applicability of the results for practical use.
Comments on the Quality of English LanguageEnglish is OK but might be refined by an English editor.
